

# Direct measurement of NO₃ reactivity in a boreal forest

Jonathan Liebmann[1], Einar Karu[1], Nicolas Sobanski[1], Jan Schuladen[1], Mikael Ehn[3], Simon Schallhart[3], Lauriane Quéléver[3], Heidi Hellen[2], Hannele Hakola[2], Thorsten Hoffmann[4], Jonathan Williams[1], Horst Fischer[1], Jos Lelieveld[1] and John N. Crowley[1]

[1]Division of Atmospheric Chemistry, Max Planck Institut für Chemie, 55128, Mainz, Germany
[2]Finnish Meteorological Institute, 00560, Helsinki, Finland
[3]Department of Physics, University of Helsinki, 00140, Helsinki, Finland
[4]Johannes Gutenberg University, 55128, Mainz, Germany

*Correspondence to:* John Crowley (john.crowley@mpic.de)

**Abstract.** We present the first direct measurements of NO₃ reactivity (or inverse lifetime, $s^{-1}$) in the Finnish boreal forest. The data were obtained during the IBAIRN campaign (Influence of Biosphere-Atmosphere Interactions on the Reactive Nitrogen budget) which took place in Hyytiälä, Finland during the summer / autumn transition in September 2016. The NO₃ reactivity was generally very high with a maximum value of 0.94 $s^{-1}$ and displayed a strong diel variation with a campaign-averaged nighttime mean value of 0.11 $s^{-1}$ compared to a daytime value of 0.04 $s^{-1}$. The highest nighttime NO₃-reactivity was accompanied by major depletion of canopy level ozone and was associated with strong temperature inversions and high levels of monoterpenes. The daytime reactivity was sufficiently large that reactions of NO₃ with organic trace gases could compete with photolysis and reaction with NO. There was no significant reduction in the measured NO₃ reactivity between the beginning and end of the campaign indicating that any seasonal reduction in canopy emissions of reactive biogenic trace gases was offset by emissions from the forest floor. Observations of biogenic hydrocarbons (BVOC) suggested a dominant role for monoterpenes in determining the NO₃ reactivity. Reactivity not accounted for by in-situ measurement of NO and BVOCs was variable across the diel cycle with, on average, ≈ 30% "missing" during nighttime and ≈ 60% missing during the day. Measurement of the NO₃ reactivity at various heights (8.5 to 25 m) both above and below the canopy, revealed a strong nighttime, vertical gradient with maximum values closest to the ground. The gradient disappeared during daytime due to efficient vertical mixing.

## 1 Introduction

Biogenic and anthropogenic volatile organic compounds (VOC) have a significant impact on air quality and human health and knowledge of their tropospheric lifetimes, determined by the oxidizing capacity of the lowermost atmosphere, is a





prerequisite to predicting future atmospheric composition and climate change (Lelieveld et al., 2008). Recent estimates (Guenther et al., 2012) suggest that about 1000 Tg of biogenic volatile organic compounds (BVOC), are emitted annually by vegetation. The boreal forest covers an area of $\approx$ 15 million $km^2$ worldwide, which is of comparable size to that of the tropical rainforest (Eerdekens et al., 2009). Forests emit large amounts of unsaturated hydrocarbons in the form of the

terpenoids such as, isoprene (2-methylbuta-1,3-diene, $C_5H_8$), monoterpenes ($C_{10}H_{16}$) and sesquiterpenes ($C_{15}H_{24}$) that have a significant impact on $HOx$ ($HO + HO_2$) and $NO_X$ ($NO+NO_2$) budgets (Hakola et al., 2003; Tarvainen et al., 2005; Holzke et al., 2006; Lappalainen et al., 2009) and formation of secondary organic particle (Hallquist et al., 2009).

Along with reaction with $O_3$, BVOCs are oxidized in the troposphere by reaction with the OH and $NO_3$ radicals. OH radical-induced oxidation taking place mainly during daytime with the $NO_3$ radical (formed by reaction of $O_3$ with $NO_2$, (R1))

accounting for the major fraction of radical-induced loss of BVOC at nighttime (Wayne et al., 1991; Atkinson, 2000; Atkinson and Arey, 2003a,   b; Brown and Stutz, 2012; Mogensen et al., 2015; Ng et al., 2016; Liebmann et al., 2017). The rapid photolysis of $NO_3$ by sunlight (R5, R6) and reaction with NO (R2) typically reduces its lifetime to a few seconds during day-time. At night-time, reaction of $NO_3$ with $NO_2$ results in thermal equilibrium between $NO_3$ and $N_2O_5$ (R3, R4).

$$NO_2 + O_3 \qquad \rightarrow NO_3 + O_2 \tag{R1}$$

$$NO_3 + NO \qquad \rightarrow 2\,NO_2 \tag{R2}$$

$$NO_2 + NO_3 + M \rightarrow N_2O_5 + M \tag{R3}$$

$$N_2O_5 + M \qquad \rightarrow NO_3 + NO_2 + M \tag{R4}$$

$$NO_3 + h\nu \qquad \rightarrow NO_2 + O \tag{R5}$$

$$NO_3 + h\nu \qquad \rightarrow NO + O_2 \tag{R6}$$

$$N_2O_5 + \text{surface} \quad \rightarrow NO_3^- \text{ (and/or ClNO}_2) \tag{R7}$$

Reactions (R1) to (R6) do not represent a reduction of $NO_X$ as no reactive nitrogen species are removed from the gas-phase. However, heterogeneous uptake of $N_2O_5$ to particles (R7) and the reaction of $NO_3$ with BVOCs (forming either $HNO_3$ or organic nitrates, see below) both result in the transfer of gas-phase $NO_X$ to particulate forms, thus reducing the rate of photochemical $O_3$ formation from $NO_2$ photolysis (Dentener and Crutzen, 1993).

In forested environments at low $NO_X$ the lifetime of $NO_3$ with respect to chemical losses during the temperate months will generally be driven by the terpenoids, the reaction proceeding via addition to the C=C double bond to form nitroxy-alkyl peroxy radicals. The peroxy radicals react further (with $HO_2$, NO, $NO_2$ or $NO_3$) to form multi-functional peroxides and organic nitrates, which can contribute to the generation and growth of secondary organic aerosols (Ehn et al., 2014; Fry et al., 2014; Ng et al., 2016; Liebmann et al., 2017) or be lost by deposition. The main processes outlining the role of $NO_3$ in

removing $NO_X$ from the atmosphere are summarized in Fig. 1. Clearly, the lifetime of $NO_3$ with respect to reaction with BVOC (the subject of this manuscript) determines the relative rate of formation of inorganic nitrate via heterogeneous processes ($HNO_3$) and organic nitrates, which have different lifetimes with respect to chemical and depositional loss and thus different efficiencies of $NO_X$ removal. It also indirectly determines the rate of generation of reactive chlorine (in the form of





ClNO$_2$) resulting from the heterogeneous reactions of N$_2$O$_5$ with chloride containing particles (Osthoff et al., 2008; Thornton et al., 2010; Phillips et al., 2012; Ammann et al., 2013; Phillips et al., 2016).

Reactivity measurements have previously been used applied to assess the overall loss rates of the OH-radical in the boreal forest and to test for closure in its budget (Hens et al., 2014). In forested environments, the measured reactivity has generally

been found to be significantly higher than that calculated from summing up reactivity due to individual reactive trace gases, (Sinha et al., 2010; Nölscher et al., 2012; Nolscher et al., 2016) resulting in an apparent "missing reactivity". In a similar vein, O$_3$ flux measurements in Californian pine forests required monoterpene emissions that were 10 times higher than measured in order to explain the O$_3$ losses (Goldstein et al., 2004). These studies argue for the presence of monoterpenes / sesquiterpenes that are not detected by standard instruments used to measure BVOC. Direct measurements of NO$_3$ reactivity

were not available until very recently (Liebmann et al., 2017) and the reactivity of NO$_3$ has traditionally been calculated from concentration measurements by assuming balanced production and loss terms (stationary state, see Sect. 3.4) or from measurements of the VOCs that contribute to its loss and the known rate constant for reaction of each VOC with NO$_3$. The first method may break down when stationary state is not achieved (Brown et al., 2003). For example, Sobanski et al. (2016b) observed much lower stationary-state loss rates of NO$_3$ compared to those calculated from measured VOC mixing

ratios in a forested /urban location and concluded that this was mainly the result of sampling from a low-lying residual layer with VOC emissions that were too close to the sampling point for NO$_3$ concentrations to achieve stationary-state. The second method relies on comprehensive measurement and accurate quantification of all VOCs that react with NO$_3$, which, in a chemically complex environment such as a forest, may not always be possible.

In this paper we describe direct, point measurements of NO$_3$ reactivity in ambient air in the boreal forest of southern Finland

and analyze the results using ancillary measurements of NO$_X$, NO$_3$, O$_3$ and biogenic hydrocarbons as well as meteorological parameters.

## 2 Measurement site and instrumentation

The IBAIRN campaign took place in September 2016 in the Boreal forest at Hyytiälä, Finland. September marks the transition from late summer to autumn at Hyytiälä, with the number of daylight hours at the site changing from ≈ 14 hours to

11.5 hours from the beginning to the end of September, with the first widespread ground frost occurring close to the end of the campaign. The relative humidity within the canopy frequently reached 100% at night-time, though there was little rainfall during the campaign. By the end of the campaign, the initially green leaves of deciduous trees had turned brown and fresh needle/leaf litter was accumulating on the forest floor. The daily profiles of temperature, relative humidity and the NO$_3$ photolysis rate constant ($J_{NO3}$) are displayed in Figure 2.



## 2.1 SMEAR II site

The measurements presented here were conducted at the "Station for Measuring Forest Ecosystem-Atmosphere Relations II" (SMEAR II) in Hyytiälä (61°51' N, 24°17' E) in southern Finland at 180 m above sea level (Hari and Kulmala, 2005). This site has been focus of intensive research investigating BVOC (Rinne et al., 2005; Holzke et al., 2006; Hakola et al., 2009;

Lappalainen et al., 2009; Aaltonen et al., 2011) and their influence on $O_3$ reactivity (Mogensen et al., 2015; Zhou et al., 2017) and OH-reactivity (Sinha et al., 2010; Nölscher et al., 2012) as well as simulations on $NO_3$ lifetimes (Hakola et al., 2003; Peräkylä et al., 2014). SMEAR II is located 49 km north-east of Tampere (pop. $\approx$ 226 000; 430 inh./km$^2$) and 88 km south-west of Jyväskylä (pop. $\approx$ 137 000; 120 inh./km$^2$). Anthropogenic influence at the site is generally low, especially when the wind comes from the sparsely populated northern sector. Operations of a sawmill, a wood mill and a pellet factory

in Korkeakoski, 5 km southeast of Hyytiälä, can result in elevated levels of monoterpenes at SMEAR II (Eerdekens et al., 2009; Liao et al., 2011; Williams et al., 2011; Hakola et al., 2012). Furthermore, pollution from forest management as well as minor influences from nearby settlements with low population densities are possible.

Figure 2 shows the local wind-speed and wind-direction at 16 m height (close to canopy top) during the campaign. The wind-rose in Fig. 3a indicates that the prevailing wind was from the NW and NE sectors ($\approx$ 60% of the time) compared to

28% from the southern sector, of which only $\approx$ 8% came from the SE. Nonetheless, two isolated plumes from Korkeakoski were evident as greatly increased values of the $NO_3$ reactivity and BVOC levels, as discussed later. In general wind speeds at 16 m height were low, favoring a stable boundary layer during night-time. Wind speed, wind direction, temperature, precipitation, relative humidity were monitored at various heights on the 128 m SMEAR II tower. Details regarding these and other supporting measurements made at this site can be found elsewhere (Hari and Kulmala, 2005; Hari et al., 2013).

The vegetation at the site consists mostly of Scots pine (*Pinus sylvestris*, >60%) with occasional Norway spruce (*Picea abies*), aspen (*Populus sp.*) and birch (*Betula sp.*). The most common vascular plants are lingonberry (*Vaccinium vitis-idea* L.), bilberry (*Vaccinium myrtillus* L.), wavy hair-grass (*Deschampsia flexuosa* (L.) Trin.) and heather (*Calluna vulgaris* (L.) Hull.). The ground is covered with common mosses as such as Schreber's big red stem moss (*Pleurozium schreberi* (Brid.) Mitt.) and a dicranium moss (*Dricanum* Hedw. sp.). The canopy height is $\approx$20 m with an average tree density of 1370 stems

(diameter at breast height > 5 cm) per hectare (Ilvesniemi et al., 2009).

## 2.2 NO$_3$ reactivity measurement

$NO_3$ reactivity was measured using an instrument that was recently described in detail by Liebmann et al. (2017). Briefly, 40 to 60 pptv of synthetically generated $NO_3$ radicals (R1) were mixed with either zero air (ZA) or ambient air in a cylindrical flow-tube thermostatted to 21 °C. After a reaction time of 10.5 s, the remaining $NO_3$ was detected by cavity-ring-down

spectroscopy (CRDS) at 662 nm. The measurement cycle was typically 400 s for synthetic air and 1200 s for ambient air, with intermittent signal zeroing (every $\approx$ 100 s) by addition of NO.





The observed loss of $NO_3$ in ambient air compared to ZA was converted to a reactivity via numerical simulation of a simple reaction scheme (Liebmann et al., 2017) using measured amounts of NO, $NO_2$ and $O_3$. The parameter obtained, $k_{OTG}$, is a loss rate constant for $NO_3$ from which contributions from NO and $NO_2$ have been removed, and thus refers to reactive loss to organic trace gases (OTG) only. Throughout the manuscript, $NO_3$ reactivity and $k_{OTG}$ are equivalent terms, with units of $s^{-1}$.

The dynamic range of the instrument was increased to 0.005 - 45 $s^{-1}$ by automated, dynamic dilution of the air sample, the limit of detection being defined by the stability of the $NO_3$ source. Online calibration of the reactivity using an NO standard was performed every 2 hours for 10 min. The uncertainty of the measurement was between 0.005 and 0.158 $s^{-1}$, depending mainly on dilution accuracy, NO levels and stability of the $NO_3$ source (Liebmann et al., 2017).

The instrument was operated in a laboratory container located in a gravel-bedded clearing in the forest. Air samples were

drawn at a flow rate of 2900 standard cubic centimeters per minute (sccm) through a 2 µm membrane filter (Pall Teflo) and 4 m of PFA tubing (6.35 mm OD) from the center of a high-flow inlet (Ø = 15 cm, flow = 10 $m^3$ $min^{-1}$) which sampled at 8 m height, 3 m above the roof of the container and circa 8 m away from the forest edge. Several instruments sampled from the high-flow and we refer to this as the "common inlet". Relative humidity and temperature were monitored in the common inlet using standard sensors ($1^{st}$ Innovative Sensor Technology, HYT939, ± 1.8% RH). Vertical profiles of $NO_3$ reactivity

(8.5 to 25 m) were measured by attaching 30 m of PFA tubing directly to the flow-tube and raising / lowering the open end (with membrane filter) using a rope-hoist attached to a 30 m tower about 5 m from the container.

## 2.3 NO, $NO_2$, $O_3$ and $NO_3$ measurements

NO was sampled from the common inlet using a modified commercial chemiluminescence detector (CLD 790 SR) based on the reaction between NO and $O_3$ (ECO Physics, Duernten, Switzerland). The detection limit for NO was 5 pptv for an

integration period of 5 s, the total uncertainty (2σ) was 20% (Li et al., 2015). Ozone was measured by two instruments based on optical absorption, both sampling from the common inlet. These were a 2B-Technology, Model 202 and a Thermo Environmental Instruments Inc., Model 49 both with detection limits of ≈ 1 ppb. The two instruments had uncertainties (provided by manufacturer) of 5% and 2%, respectively. Agreement between the two $O_3$ measurements was excellent (slope = 1.000 ± 0.001, offset of -0.21 ppbv, $R^2$ = 0.98). Vertical profiles in $O_3$ (up to 125 m) were made using a TEI 49 C analyzer

sampling from inlets at various heights on a tower located 130 m NNW of the measurement container. $NO_2$ and $NO_3$ were measured from the common inlet using a multi-channel, Thermal Dissociation-Cavity Ring Down spectrometer (TD-CRDS) recently described in detail by (Sobanski et al., 2016a). $NO_3$ radicals were detected at 662 nm with a detection limit of 1.3 pptv (1 min averaging) and an uncertainty of 25%. $NO_2$ was detected at 405 nm with an uncertainty of 6% and a detection limit of 60 pptv (1 min averaging).

## 30 2.4 VOC measurements

Three different instruments were used to monitor VOCs. These were 1) a gas chromatograph equipped with an atomic emission detector (GC-AED) which sampled from the common inlet, 2) a thermal desorption gas chromatograph with mass



spectrometric detection (GC-MS) sampling about 1.5 m above the ground and ≈ 10 m distant from the reactivity measurements and 3) and a proton transfer reaction time of flight mass spectrometer (PTR-TOF-MS) located about 170 m away in dense forest and sampling at a height of ≈ 2.5 m above the ground.

### 2.4.1 GC-AED

The GC-AED consisted of a cryogenic pre-concentrator coupled to an Agilent 7890B GC and an atomic emission detector (JAS AEDIII, Moers, Germany). The GC-AED sampled air through a 15 m long, ½" (1.27 cm) outer diameter PFA Teflon tube (flowrate = 20 L min$^{-1}$, transmission time 3.3 s) which was heated to ≈ 10°C above ambient. The instrument was calibrated *in-situ* with an 84-component gravimetrically prepared gas-phase calibration reference standard with a stated accuracy of better than ±5% (Apel-Riemer Environmental, Inc., Florida, USA). The average total uncertainty of the species

measured from repeated calibration standard measurements combined with the flow measurements and calibration standard uncertainty was calculated 14%. α-pinene, Δ-3-carene, β-pinene, camphene and d-limonene were all calibrated individually. Detection limits for the monoterpene species were 1.0, 0.9, 0.4, 0.5 and 0.3 pptv respectively. As this is the first deployment of this instrument, more details are provided in the supplementary information: A full description will be the subject of an upcoming publication.

### 2.4.2 GC-MS

The GC-MS was located in a container in a gravel-bedded clearing about 4 meters from the edge of the forest and ≈30 m away from the common inlet. Air samples were taken every other hour (30 min sampling time) at a height of 1.5 m by drawing air at 1 L min$^{-1}$ through a 1m long fluorinated ethylene propylene (FEP) inlet (id. 1/8 inch). Ozone was removed via a heated (120°C) stainless steel tube (Hellén et al., 2012). VOCs were collected from a 40 ml min$^{-1}$ subsample flow into the

cold trap (Tenax TA/ Carbopack B) of the thermal desorption unit (TurboMatrix, 650, Perkin-Elmer) connected to a gas chromatograph (Clarus 680, Perkin-Elmer) with HP-5 column (60m, id. 0.25mm, film thickness 1 μm) coupled to a mass spectrometer (Clarus SQ 8 T, Perkin-Elmer). The instrument was used for measurements of isoprene, monoterpenes and aromatic hydrocarbons and was calibrated for all individual compounds using liquid standards in methanol solutions, which were injected into the Tenax TA/Carbopack B adsorbent tubes and analyzed with the same method as the air samples.

Detection limits for monoterpenes (α-pinene, camphene, β-pinene, 3Δ-carene, myrcene, p-cymene, limonene, 1,8-cineol, terpinolene) were 0.2-1.2 pptv and for β-caryophyllene 0.8 pptv. The average total uncertainty (10% for all monoterpenes and β-caryophyllene) was calculated from the repeatability of the calibrations, uncertainty of the standard preparation and the uncertainty in the sampling flow.



### 2.4.3 PTR-TOF-MS

The PTR-TOF-MS (PTR-TOF 8000, Ionicon Analytic GmbH) measures whole VOC spectra in real time (Jordan et al., 2009; Graus et al., 2010) with mass resolution of 4500 (full width at half maximum). The instrument was located in the main cottage, approximately 170 m away from the common inlet. Ambient air was sampled from 2.5 m above the ground, using a

3.5 m long (4 mm inner diameter) PTFE sampling air at 20 L min$^{-1}$. A sub-sample flow of 1 L min$^{-1}$ was passed via 10 cm of PTFE tubing (1.6 mm i.d.), via a three-way valve and 15 cm of PEEK tubing (1 mm i.d.) to the PTR-TOF-MS. The raw data was collected with a 10 s resolution. The instrument measured total monoterpenes at $m/z = 137$ and isoprene at $m/z = 69$, which were calibrated with a gas standard (Apel Riemer Environmental Inc., USA) containing isoprene and α-pinene. The calibration set up and routine are described in detail in Schallhart et al. (2016). The campaign average limit of detection LOD

(3 σ, 10 minute time resolution) was 5.5 pptv and 3.2 pptv for isoprene and monoterpenes, respectively.

### 3 Results and discussion

NO$_X$ mixing ratios were generally low during the campaign with NO$_2$ between 0.1 to 1.84 ppbv with a campaign average of 0.32 ppbv showing little variation across the diel cycle. The mean day-time NO mixing ratio was 43 pptv while nighttime NO was close to or below the limit of detection (≈5 pptv) and its contribution to the loss of NO$_3$ was generally insignificant

(see below). Ozone mixing ratios showed large day / night differences with daily maxima between 30 and 40 ppbv, but nighttime values as low as 5-10 ppbv. Possible reasons for the large changes in O$_3$ across the diel cycle are addressed in section 3.1.

### 3.1 NO$_3$ reactivity and nighttime loss of O$_3$

NO$_3$ reactivity was measured from 5$^{th}$ of September 12:00 UTC to the 22$^{nd}$ of September 05:30 UTC; the 1-minute averaged

time series of $k_{OTG}$ is displayed in Figure 2. The overall uncertainty in $k_{OTG}$ is given by the green, shaded region. NO$_3$ photolysis and reaction with NO result in concentrations that are generally below the detection limit of modern instruments during daytime and steady-state calculations of NO$_3$ reactivity are lower limit estimates. In contrast, our direct approach allows us to derive and analyze daytime values of $k_{OTG}$ as long as NO$_X$ measurements are available (see above). Figure 2 indicates that, in general, the NO$_3$ reactivity was highest at nighttime, the maximum observed values in $k_{OTG}$ was 0.94 s$^{-1}$, (at

21:00 UTC on 9$^{th}$ Sept) implying a lifetime of just 1 s and a very reactive air mass at this time. The mean nighttime value of $k_{OTG}$ was ≈ a factor ten lower at 0.11 s$^{-1}$, the daytime mean even lower at 0.04 s$^{-1}$. Broadly speaking, the nighttime NO$_3$ lifetimes during IBAIRN were very short (≈ 10 s on average) compared to previous indirect, ground level measurements in other locations where several groups have reported lifetimes of hundreds to thousands of seconds (Heintz et al., 1996; Allan et al., 1999; Geyer et al., 2001; Aldener et al., 2006; Ambrose et al., 2007; Brown et al., 2009; Crowley et al., 2010; Crowley

et al., 2011; Sobanski et al., 2016b).




Figure 3b illustrates the dependence of $k_{OTG}$ on wind direction. Air masses from the northern sector were generally associated with lower reactivity ($<0.2$ s$^{-1}$) whereas all incidents of reactivity larger than 0.3 s$^{-1}$ were associated with air masses from the SE sector. Enhanced reactivity from the SE may be caused by emissions from the sawmill at Korkeakoski (Eerdekens et al., 2009), or a local wood-shed storing freshly cut timber about 100 m distant from the containers.

Additionally, the lower than average wind-speeds associated with air masses from the SE, may have reduced the rate of exchange between the nocturnal boundary layer and above canopy air, effectively trapping ground-level emissions into a shallow boundary layer. Emissions from the sawmill reaching the site on the night from the 9$^{th}$ to the 10$^{th}$ of September provided a useful test of our method at high reactivity.

In order to examine the difference in daytime and nighttime NO$_3$ reactivity and also to explain the large nighttime variability

in $k_{OTG}$ we categorize the nights into three broad types: 1) nights where the NO$_3$-reactivity was greatly increased compared to the previous or following day, 2) nights with comparable (usually low) daytime and nighttime NO$_3$-reactivity, and 3) events with unusually high NO$_3$-reactivity. Figure 4 zooms in on $k_{OTG}$ data over a five day/night period (5$^{th}$-10$^{th}$ Sept) in which all three types are represented. It also plots the temperature at different heights as well as the RH and O$_3$ measured in the common inlet at 8.5 m height.

**3.1.1 Type 1 and type 2 nights**

Within this 5-day period, the nights on which the reactivity was high relative to the day (type 1) are the 5$^{th}$-6$^{th}$ and 8$^{th}$-9$^{th}$. These nights are characterized by large depletion in O$_3$, a significant temperature inversion of 5 - 7 °C between heights of 8 and 128 m, and a relative humidity of 100% directly after sunset. In contrast, two interspersed nights with comparable reactivity to daytime values (6$^{th}$-7$^{th}$, 7$^{th}$-8$^{th}$, type 2) display much weaker (if any) nighttime loss of O$_3$ compared to levels

during the previous day, no significant temperature inversion and a relative humidity less than 100%. The observations within this period can be extended to all campaign days. Figure 5 presents the diel cycle of $k_{OTG}$ and O$_3$ mixing ratios separated into nights of type 1 (with temperature inversion) and type 2 (no temperature inversion). The shaded regions represent the variability of the measured values. During 24 hour periods in which the night was characterized by strong temperature inversion (upper panel in Fig. 5), the O$_3$ mixing ratios display a large diel variation, with a maximum of 35 ppbv

at about 13:00 UTC dropping rapidly to a minimum of $\approx$ 13 ppb between midnight and sunrise at around 05:00 UTC. The O$_3$ mixing ratio shows an inverse diel profile to the NO$_3$ reactivity raising the possibility that the rapid loss of ozone is linked to high NO$_3$ reactivity; the large values of $k_{OTG}$ and rapid O$_3$ depletion observed on nights with a significant temperature inversion are clear indicators that nighttime boundary layer dynamics plays a key role in controlling both the NO$_3$ reactivity and O$_3$ loss. A strong nocturnal temperature inversion will weaken the mixing within or ventilation of the lowermost

boundary layer causing a build-up of reactive, biogenic emissions in the lower layer that remove NO$_3$, and also prevent down-mixing of drier, O$_3$-rich air leading to the apparent higher loss rate of O$_3$ and higher relative humidity. The strong anti-correlation between $k_{OTG}$ and O$_3$ may also provide a clue to the origin of the O$_3$ loss. Whilst the generally high NO$_3$ reactivity can, to a large extent, be explained by the presence of reactive trace gases (see section 3.2), the precipitous loss of



$O_3$ on several nights when $k_{OTG}$ was high (see Fig. 2 and Fig. 4) may have components of both dry deposition and gas-phase reactions.

The campaign averaged, diel variation of ozone at different heights (4 m to 125 m) as measured at the SMEAR II tower (Fig. S1 of the supplementary information) indicates that the most rapid losses of ozone are at the lowest heights, around and below the canopy. Ozone is generally removed from the lower troposphere by both stomatal and non-stomatal deposition, the latter involving loss to surfaces and soil. The reactive, gas-phase loss mechanisms of $O_3$ and $NO_3$ are in some ways similar, as both react with NO to form $NO_2$, or with unsaturated VOCs by addition to the double bond. We estimated the loss rate constant for $O_3$ due to its reaction with terpenes using approximate ambient mixing ratios from 20:00 to 00:00 UTC on the 20[th] of september for $d$-limonene (20 pptv), α-pinene (400 pptv), Δ-carene (100 pptv) and β-pinene (100 pptv) and using literature rate constants for the $O_3$ + terpene reactions. The calculated $O_3$ loss (only 2% from 20:00 to 00:00 UTC) is clearly insufficient to explain the IBAIRN observations. We also note that the presence of high concentrations of terpenes when the site was impacted by the Korkeakoski sawmill resulted in the largest $NO_3$ reactivity observed, but did not lead to large $O_3$ losses (Fig. 4). As leaf stomata are closed during night-time, the decrease in $O_3$ can be attributed either to non-stomatal deposition or chemical sinks due to reaction with reactive biogenic trace gases (but not the measured monoterpenes) and NO. Previous studies of $O_3$ loss in forests have highlighted the potential role of unidentified, reactive organic compounds (Kurpius and Goldstein, 2003; Goldstein et al., 2004; Holzinger et al., 2006; Rannik et al., 2012). In contrast to monoterpenes, that react only slowly with $O_3$ (rate constants are ≈ $10^{-16}$-$10^{-17}$ cm$^3$ molecule$^{-1}$ s$^{-1}$, IUPAC (2017), sesquiterpenes (e.g. β-caryophyllene) react rapidly, in this case with a rate coefficient of $k_{O3} = 1.2 \times 10^{-14}$ cm$^3$ molecule$^{-1}$ s$^{-1}$ (IUPAC, 2017). The presence of sesquiterpenes would therefore provide an explanation for the observations of high $NO_3$ reactivity and rapid $O_3$ loss. We examine the potential role of sesquiterpenes in more detail in section 3.2 where the contribution of measured terpenoids to $NO_3$ reactivity is discussed. We also note that recent modeling studies using Hyytiälä data (Chen et al., 2017; Zhou et al., 2017) conform that $O_3$ depletion events are associated with formation of shallow boundary layer and high relative humidity. Zhou et al. (2017) conclude that chemical reaction plays only a minor role for ozone loss processes during night, which was suggested to be dominated by deposition to wet surfaces at relative humidity >70%, in accord with laboratory investigations (Sun et al., 2016).

### 3.1.2 Type 3 nights

The period between the evening and midnight on the 9[th] of September is an example of the third type of night, with extremely high $NO_3$ reactivity, which was not accompanied by significant $O_3$ depletion, temperature inversion or RH of 100%. The apparently anomalously high reactivity on this night can be traced back to a change in wind direction, which swept from easterly to southerly in this period, bringing air that was impacted by monoterpene emissions from the sawmill in Korkeakoski. High mixing ratios of terpenoids in airmasses that have passed over the sawmill have been documented frequently (Eerdekens et al., 2009; Sinha et al., 2010; Liao and Dal Maso, 2011; Hakola et al., 2012; Nölscher et al., 2012). Other occurrences of sawmill contaminated air during IBAIRN were on the 10[th] of September from 18:40 to 19:00 UTC and





on the $14^{th}$ of September from 06:30 to 08:00 UTC, HYSPLIT back-trajectories (GDAS global, 0.5°), indicating that the air mass passed over Korkeakoski ≈ 0.5 hours prior to reaching the SMEAR II site.

**3.2 Comparison of $k_{OTG}$ with $NO_3$ reactivity derived from VOC measurements**

In this section we compare $k_{OTG}$ with $NO_3$ reactivity calculated from ambient VOC mixing ratios. During IBAIRN, three instruments (GC-MS, GC-AED, PTR-TOF) measuring VOCs were deployed (see section 2.5 for details). As the PTR-TOF reports only a summed mixing ratio of all monoterpenes, ΣMT(PTR-TOF), we first generated an equivalent parameter for the two GCs, ΣMT(GC-MS) and ΣMT(GC-AED). For the GC-MS, α-pinene, β-pinene, Δ-carene, *d*-limonene, camphene, myrcene and terpinolene were considered whereas for the GC-AED, α-pinene, β-pinene, Δ-carene, camphene and *d*-limonene were taken into account. The ΣMT data are displayed as a time series in Fig. 6, which indicates large differences between the three measurements as highlighted by the histograms in Fig. S2 of the Supplementary information). While the ΣMT(GC-AED) and ΣMT(PTR-TOF) data are in reasonable agreement, especially when mixing ratios were large, the values reported by the GC-MS are consistently and significantly lower (factor 2 to >10) than those of the others instruments. The time dependent variability in the differences in ΣMT reported by the GC-MS, GC-AED and PTR-TOF is a strong indication that the cause is most likely related to instrument location and inhomogeneity in terpene emissions within the forest. Whilst the GC-AED sampled from the common inlet at 8.5 m height, which was also used for the $NO_3$ reactivity measurements, the inlet of the GC-MS was ≈ 10 m distant and sampled 1.5 m above the gravel covered clearing, very close to the side of the container which housed the instrument. The PTR-TOF-MS was located roughly 170 m away in a wooden cottage directly surrounded by dense forest and sampled close to the forest floor at a height of ≈ 1.5 m. With very low within-canopy wind-speeds, especially during nighttime, both horizontal as well as vertical mixing in the forest and in the clearing are weak so that each VOC-measurement may, to some extent, reflect the mixture and total amount of BVOCs that are very locally emitted. This aspect was examined by comparing individual monoterpenes measured by the GC-MS and the GC-AED. The results, presented as correlation plots for 4 monoterpenes in Fig. S3 of the supplementary information, show that the monoterpene ratios measured by the two instruments (GC-AED / GC-MS), were variable with values of 1.69±0.06 for α-pinene, 2.51 ± 0.09 for β-pinene, 4.29 ± 0.21 for Δ-carene and 0.45 ± 0.03 for d-limonene. A similar picture emerges for isoprene, for which the GC-AED measured mixing ratios that were a factor 2-5 larger than measured by the GC-MS. The variable relative concentrations of monoterpenes reported by each instrument is further evidence of the inhomogeneity of emissions within the forest and also the influence of different tree chemotypes within single tree-families in Hyytiälä, which can exhibit vastly different emission rates of the various monoterpenes (Bäck et al., 2012; Yassaa et al., 2012).

For the purpose of comparing our point measurements of $k_{OTG}$ with $NO_3$-reactivity calculated from BVOC measurements, we restricted our analysis to the data set obtained by the GC-AED, which sampled from the same inlet. Nonetheless, when comparing measured $NO_3$ mixing ratios with those calculated from $NO_3$-reactivity and its production term (see section 3.4) we use both GC-based datasets.





The loss rate constant, $k_{OTG}$, represents chemical reactions of [NO$_3$] with all organic trace gases present and can be compared to the loss rate constant ($k_{GC\text{-}AED}$) obtained from the concentrations of VOCs in the same air mass as measured by the GC-AED, and the rate coefficient for reaction with NO$_3$ :

$$k_{GC-AED} = \sum k_i[C_i] \, , \tag{1}$$

A difference in the values of $k_{OTG}$ and $k_{GC\text{-}AED}$ is defined as missing reactivity (s$^{-1}$):

$$\text{missing reactivity} = k_{OTG} - k_{GC\text{-}AED} \tag{2}$$

Where [C$_i$] is the measured VOC concentration and $k_i$ the corresponding rate constant. The rate constants used in these calculations of $k_{GC\text{-}AED}$ were taken from the IUPAC evaluation (IUPAC, 2017). Figure 7 (lower panel) shows the concentrations of the monoterpenes as measured by the GC-AED. The dominant monoterpene was α-pinene followed by Δ-carene, β-pinene, *d*-limonene and camphene. The GC-AED also detected myrcene and linaool and some other terpenes but the very low mixing ratios meant that none of them contributed significantly to NO$_3$ loss.

In Fig. 7 (upper panel) we overlay the time series of $k_{OTG}$ and $k_{GC\text{-}AED}$. For clarity of presentation we have omitted to plot the overall uncertainty of each measurement. This was calculated as described previously (Liebmann et al., 2017) and is plotted in Fig. S5 of the supplementary information. The uncertainty associated with $k_{GC\text{-}AED}$ was calculated by propagating uncertainty in the mixing ratios of the individual terpenes (14%, mainly resulting from on the uncertainty in the calibration standard and the calibration reproducibility) and assuming 15% uncertainty in the rate coefficients for reactions of NO$_3$ with each terpene. The values of $k_{OTG}$ and $k_{GC\text{-}AED}$ do not agree within their combined uncertainties, indicating that the missing reactivity calculated in equation (3) is statistically significant. Figure 8 plots the time series of the fractional contribution to $k_{GC\text{-}AED}$ made by monoterpenes detected by the GC-AED. The GC-AED derived NO$_3$-reactivity is dominated by α-pinene and Δ-carene and to a lesser extent *d*-limonene, with minor contributions from β-pinene, camphene and isoprene.

In Figure 9 we plot the diel profiles of $k_{OTG}$, $k_{GC\text{-}AED}$ (s$^{-1}$) averaged for the whole campaign. To do this, we interpolated the values of $k_{OTG}$, obtained with 60 s time resolution averaged to 900 s data onto the low-time resolution (≈ 60 min) GC-AED dataset. In the lower panels of Fig. 9 we plot two separate diel profiles, separating the data into nights with (middle panel) and without (lower panel) strong temperature inversion. The missing reactivity (in s$^{-1}$) across the entire diel profile is between ≈ 0.02 and 0.07, the larger missing reactivity encountered during nighttime. In contrast, the fraction of missing reactivity within the campaign averaged diel cycle was observed during daytime (≈ 60%), with only 30% missing at nighttime. The lowermost panel of Fig. 9 highlights the fact that $k_{OTG}$ was lower during campaign day / night periods with no temperature inversion and shows that it is roughly constant across the diel cycle. Likewise, the diel cycle in the reactivity attributed to the monoterpenes is also constant, with a missing reactivity of between 0.02 and 0.04 s$^{-1}$. A different picture emerges for the diel cycle considering only the days / nights with strong temperature inversion. On average, we see a much higher nighttime reactivity, which is tracked in its diel profile by that calculated from the measured monoterpenes. In this case, the missing reactivity is generally higher and more variable, with values between 0 and 0.1 s$^{-1}$.





Although statistically significant, the fraction of reactivity missing is much smaller than that reported for OH at this site (Nölscher et al., 2012) whereby up to 90% of the observed reactivity was unaccounted for when the forest was under stress due to high temperatures. For OH, the fractional missing reactivity was also greatest when the overall reactivity was high, which is in contrast with the situation for $NO_3$ where missing reactivity was highest when the overall reactivity was low (i.e.

during daytime). The OH radical reacts with most hydrocarbons and many inorganic trace gases and may be considered unselective in its reactivity, whereas $NO_3$ is a more specific oxidant of VOCs, its reactions in the forest dominated by addition to unsaturated VOCs or reaction with NO.

As the nighttime mixing ratios of NO were low (< 5 pptv apart from the night $20^{th}$-$21^{st}$ Sept when a mixing ratio of ≈ 25 pptv was measured), its contribution to the overall nighttime loss of $NO_3$ was insignificant. Figure S6 of the supplementary

information indicates that, averaged over the entire campaign, NO accounted for less than 2% of the reactive loss of $NO_3$ at night. A different picture emerges for daytime, for which the campaign averaged contribution of NO to the overall chemical reactivity of $NO_3$ peaked at 40% at about 10:00 UTC. However, even during daytime, the average missing reactivity of 0.025 $s^{-1}$ (Fig. 9) would require an extra 40 pptv of NO to account for it, which is clearly not within the total uncertainty of the NO measurement.

A more plausible explanation for the missing $NO_3$ reactivity is incomplete detection of all reactive BVOCs by the GC-AED, which does not report mixing ratios of some hydrocarbons such as e.g. 2-methyl-3-buten-2-ol, p-cymene, 1,8 cineol, which the GC-MS showed to be present. The GC-MS mixing ratios of these species (which react slowly with $NO_3$) were however too low for them to contribute significantly, even taking into account the potentially larger concentrations at the common inlet.

We also consider the potential role of sesquiterpenes. Mixing ratios of β-caryophyllene reported by the GC-MS were generally low, with a maximum value of 25 pptv. However, the rate coefficient reported (Shu and Atkinson, 1995) for the reaction of $NO_3$ with β-caryophyllene is large ($1.9 \times 10^{-11}$ $cm^3$ $molecule^{-1}$ $s^{-1}$) and sesquiterpenes at levels of 10s of pptv can contribute significantly to $NO_3$ loss rates. Like monoterpenes, the emissions of sesquiterpenes are driven by temperature, with tree emissions most important during the hottest months (Duhl et al., 2008). Whilst previous studies at this site (Hakola

et al., 2006) found no correlation between the β-caryophyllene and monoterpene emissions of an enclosed Scots Pine branch, we find that β-caryophyllene mixing ratios (reported by the GC-MS) are correlated with those of several monoterpenes measured by the same instrument. This is illustrated in Fig. S4 of the supplementary information which indicates β-caryophyllene / monoterpene ratios (α-pinene β-pinene and Δ-carene) of $0.061 \pm 0.002$ ($R^2$ 0.86), $0.294 \pm 0.011$ ($R^2$ 0.86), and $0.181 \pm 0.007$ ($R^2$ 0.84), respectively. As the monoterpenes and sesquiterpenes have very different lifetimes with respect

to chemical loss, we have excluded the sawmill impacted data (red data point) as sesquiterpenes are unlikely to survive the ≈ 0.5 hrs. transport time from Korkeakoski due to their rapid reaction with $O_3$. The high levels of β-caryophyllene measured may indicate that the source during IBAIRN is unlikely to be Scots pine, the emissions from which are strongly temperature dependent during the summer months but low and independent of temperature in September (Hakola et al., 2006).





A rough estimate of the β-caryophyllene mixing ratio at the common inlet may be obtained from the GC-AED measurement of α-pinene and the α-pinene / β-caryophyllene ratios measured by the GC-MS (see above). The resulting β-caryophyllene mixing ratios lie between 10 and 60 pptv, which, based on a rate constant of $1.9 \times 10^{-11}$ cm$^3$ molecule$^{-1}$ s$^{-1}$, results in a contribution to NO$_3$ reactivity of up to 0.03 s$^{-1}$. As β-caryophyllene emissions from pine tree needles reveals a strong

temperature dependence (Hakola et al., 2006) it seems unlikely that this is an important source of β-caryophyllene during the relatively cold September nights of the IBAIRN campaign and its emissions from other sources, especially those at ground level including soil may be more important (Insam and Seewald, 2010; Penuelas et al., 2014).

In summary, the BVOC measurements indicate that NO$_3$ reactivity in this boreal environment is dominated by reaction with monoterpenes with, on average, 70% of the reactivity during night-time and 40% of the reactivity during day-time explained

by α- and β-pinene, Δ-carene, limonene and camphene. Unidentified monoterpenes / sesquiterpenes are likely to account for a significant fraction of the missing reactivity.

### 3.3 NO$_3$ levels: Measurements versus calculations using production and loss terms

Previous estimates of NO$_3$-reactivity (often reported as its inverse lifetime) have relied on NO$_3$ concentration measurements and the assumption that the production and loss of NO$_3$ are in stationary state. By combining $k_{OTG}$ and other loss processes

such as photolysis and reaction with NO with the NO$_3$ production term, we can also calculate the NO$_3$ concentration:

$$[NO_3]_{ss} = \frac{NO_3 \text{ production rate}}{NO_3 \text{ loss rate}} = \frac{[O_3][NO_2]k_1}{([k_{RTG}]+[J_{NO3}]+[NO]k_2)} \qquad (3)$$

Where $k_2$ is the rate constant (cm$^3$ molecule$^{-1}$ s$^{-1}$) for reaction of NO$_3$ with NO and $J_{NO3}$ is its photolysis rate constant (s$^{-1}$). $J_{NO3}$ was calculated from actinic flux measurements (spectral radiometer, Metcon GmbH, Meusel et al. (2016)) and NO$_3$ cross sections / quantum yields from an evaluation (Burkholder et al., 2016).

Figure 10 plots the time series of measured NO$_3$ mixing ratios (1 min averages, blue lines) for the entire campaign, which indicates that NO$_3$ was always below the detection limit of 1.3 pptv, which is defined by variation in the zero-signal rather than random noise (Sobanski et al., 2016a). The fact that the measured NO$_3$ mixing ratios are slightly negative (by ≈ 0.2 pptv) is due to a few percent NO$_2$ contamination of the NO-sample used to zero the NO$_3$ signal. We also plot the time series (black line) of the stationary state NO$_3$ mixing ratios, $[NO_3]_{ss}$, calculated according to (Eqn. 3). The low NO$_X$ levels and

moderate O$_3$ levels combine to result in a weak production rate for NO$_3$ of less than 0.03 pptv s$^{-1}$ for the entire campaign resulting in predicted levels of $[NO_3]_{ss}$ of less than 0.2 pptv. On two nights, higher mixing ratios close to 1 pptv (nights of 6-7th and 10-11th) are predicted, a result of elevated production rates due to higher NO$_2$ levels.

The advantages of directly measured $k_{OTG}$ rather than reactivity calculations based on measurements of reactive trace gases is illustrated by plotting the predicted NO$_3$ levels based on the reactive hydrocarbons reported by the GC-AED and GC-MS, i.e.

use of $k_{GC-MS}$ and $k_{GC-AED}$ rather than $k_{OTG}$. Use of the GC-MS data, which reported the lowest levels of biogenic hydrocarbons, would lead to the prediction of measureable amounts (up to 4 pptv) of NO$_3$ on several nights, contradicting our NO$_3$ measurements and previous reports (Rinne et al., 2012) of very low NO$_3$ levels at this site.





### 3.4 Vertical gradient in NO₃ reactivity

Both column- and point measurements of tropospheric $NO_3$ indicate a strong vertical gradient in its mixing ratio with significantly elevated levels aloft (Aliwell and Jones, 1998; Allan et al., 2002; von Friedeburg et al., 2002; Stutz et al., 2004; Brown et al., 2007a; Brown et al., 2007b; Brown and Stutz, 2012). The $NO_3$ gradient is the result of lower production rates

close to the ground, where $O_3$ levels are depleted due to deposition and also lower loss rates aloft as the concentration of reactive traces gases from ground level emissions decreases with altitude. High resolution data (Brown et al., 2007b) indicate that the largest gradient in $NO_3$ concentration is often found in the lowermost 50 m. Night-time monoterpene mixing ratios in forested, boreal regions have been found to display a vertical gradient, with highest mixing ratios at lower levels (Holzinger et al., 2005; Rinne et al., 2005; Eerdekens et al., 2009). This is a result of direct emissions of e.g. monoterpenes from the

trees at canopy level and emissions of monoterpenes and sesquiterpenes from rotting leaf-litter into a shallow, stratified boundary layer, suggesting that reactive species close to the ground will dominate in controlling the $NO_3$ lifetime and thus mixing ratio (Aaltonen et al., 2011). We explored this by making measurements of $k_{OTG}$ at various heights above ground, including measurements from a few meters below the canopy, to a few meters above the tree tops. Altogether we recorded 14 vertical profiles on the 18.09.2016, 5 obtained during the daytime (10:15 to 15:15 UTC) and 9 obtained at nighttime

(16:00-24:00 UTC).

Figure 11 displays the averaged nighttime and daytime values of $k_{OTG}$ recorded at 8.5, 12.0, 17.0, 22.0, 27.0 m. The total time to take a single profile was < 15 mins. During the day (black data points), we find no significant vertical gradient in $NO_3$ reactivity, which was roughly constant at ≈ 0.03 s⁻¹. In contrast, the average nighttime vertical profile (red data points) reveals a strong gradient in $k_{OTG}$ with the highest values slightly below canopy height (8.5 to 12.5 m) with a rapid decrease

above. At 20 m and above, daytime and nighttime values of $k_{OTG}$ were comparable. These observations are qualitatively consistent with gradients in monoterpene mixing ratios in this forest (Rinne et al., 2005) and with the conclusion of Mogensen et al. (2015) who considered $NO_3$ reactions with monoterpenes and sesquiterpenes emitted from Scots Pines at canopy height for the exceptionally warm summer of 2010. The modelled, nighttime vertical gradient in $NO_3$ described by Mogensen et al. (2015) displays a maximum at 12 m but differs from the measured one from IBAIRN in that lower reactivity

was modelled at the lowest heights, which may be expected as the model considered only emissions of reactive BVOCs from trees and not from ground sources. In contrast to the vertical gradient measured during IBAIRN, the modeled $NO_3$ reactivity showed highest values during day-time, coincident with the maximum NO mixing ratio (Mogensen et al., 2015) but was generally lower than our measured values. Mogensen et al. (2015) indicate that the model is likely to underestimate the $NO_3$-reactivity due to compounds that cannot be measured by GC-MS as well as by unknown products of their oxidation.

The increase in $k_{OTG}$ below the canopy may be caused by ground level emissions of reactive trace gases from tree- and plant-debris or other fauna (mosses, lichens) at forest-floor level. α-pinene and $\Delta^3$-carene, emissions from ground level may vary with litter quality and quantity, soil microbial activity and physiological stages of plants (Warneke et al., 1999; Insam and Seewald, 2010; Aaltonen et al., 2011; Penuelas et al., 2014). Previous work in the tropical forest in has indicated that





sesquiterpenes concentrations can peak at ground level rather than within the canopy (Jardine et al., 2011), though the applicability of this result to the boreal forest is unclear.

We conclude that high rates of emission of reactive gases into the stratified nocturnal boundary layer along with ventilation and dilution above canopy height result in the strong nocturnal gradients in $NO_3$ reactivity. During the daytime, efficient

turbulent mixing removes the gradient. We did not obtain a vertical profile of $k_{OTG}$ on a night when the temperature inversion was absent, but expect this to be significantly weaker, as is the gradient in $O_3$ on such nights.

### 3.5 High $NO_3$ reactivity and its contribution to $NO_X$ loss

The high reactivity of $NO_3$ towards organic trace gases in the boreal environment means that other loss processes, including formation of $N_2O_5$ or reaction with NO are suppressed. To a first approximation we can assume that, at nighttime, in the

absence of NO and sunlight, each $NO_3$ radical formed in the reaction of $NO_2$ with $O_3$ will react with a biogenic hydrocarbon, resulting in formation of an organic nitrate at a yield of between 20 and 100%, depending on the identity of the organic reactant (Ng et al., 2017). The large values for $k_{OTG}$ obtained during the day means that a significant fraction of the $NO_3$ formed can be converted to organic nitrates rather than result in re-formation of $NO_2$ via reaction with NO or photolysis. The fraction, $f$, of $NO_3$ that will react with organic trace gases is given by:

$$f = \frac{k_{OTG}}{([k_{OTG}]+[J_{NO3}]+[NO]k_2)} \qquad (4)$$

Figure 12 illustrates the time series (upper plot) and the campaign averaged diel cycle (lower plot) for $f$ which varies between $\approx 0.1$ and 0.4 at the peak of the actinic flux, the variation caused largely by day-to-day variability in insolation. As the spectral radiometer was located at 35 m height, $J_{NO3}$ will be slightly overestimated around midday as light levels within the canopy are lower. The overestimation will be magnified during the early morning and late afternoon when the forest is in

shade at lower levels but the spectral-radiometer is not. The daytime values for $f$ are thus lower limits. With typical daytime NO levels of 50-100 pptv, the term $[NO]k_2$ contributes $\approx 0.03$-$0.06$ s$^{-1}$ to $NO_3$ loss, whereas $J_{NO3}$ has maxima of close to 0.1 s$^{-1}$ each day. For comparison daytime values of $k_{OTG}$ of $\approx 0.05$ s$^{-1}$ were often observed (Fig. 2).

The diel cycle for $f$ shows that even at the peak of the actinic flux, on average circa 20% of the $NO_3$ formed will react with an organic trace gas rather than be photolysed or react with NO in this environment. This implies that, in the summer-autumn

boreal forest, $NO_3$ reactions may represent a significant loss of $NO_X$ not only during the nighttime but over the full diel cycle, with a significant enhancement in the daytime production of alkyl nitrates, generally assumed to proceed only via reactions of organic peroxy radicals with NO.

### 3.6 Conclusions

The first direct measurements of $NO_3$ reactivity to organic trace gases ($k_{OTG}$) in the boreal forest indicate that $NO_3$ is very

short lived in this environment with lifetimes generally less than 10s, mainly due to reaction with monoterpenes. The highest $NO_3$ reactivities were encountered during nights with strong temperature inversions, relative humidity of 100% and were





accompanied by rapid $O_3$ depletion, together highlighting an important role for nocturnal boundary layer dynamics in controlling canopy-level $NO_3$-reactivity. The daytime reactivity was sufficiently large that reactions of $NO_3$ with organic trace gases could compete with photolysis and reaction with NO, so that $NO_3$-induced losses of $NO_X$ and formation of organic nitrates was significant. Measurements of the vertical profile in $NO_3$ reactivity indicate a strong gradient during

nighttime, with the highest reactivity observed below canopy height highlighting a potential role for emissions of reactive trace gases from the forest floor. The hydrocarbons measured did not fully account for the observed $NO_3$ reactivity, indicating a role for unsaturated organic trace gases that were not identified, with a likely role of sesquiterpenes.

Acknowledgements: We are grateful to ENVRIplus for partial financial support of the IBAIRN campaign. We also thank
Uwe Parchatka for provision of the NO dataset.

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



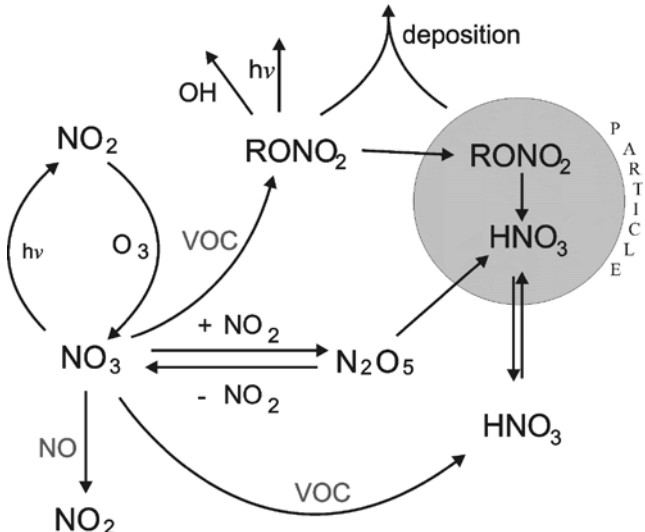

**Figure 1: Gas-phase formation and loss of tropospheric NO$_3$ indicating processes which transfer reactive nitrogen to the particulate phase. RONO$_2$ are alkyl nitrates. VOC = volatile organic compound.**




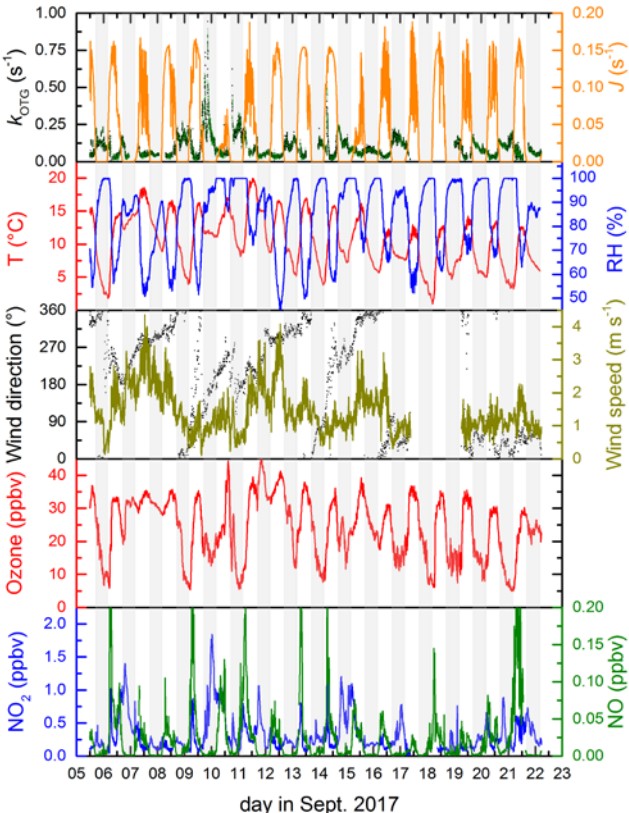

**Figure 2: Overview of measurements during IBAIRN. The grey shaded regions represent night-time. The uncertainty in $k_{OTG}$ is given by the green shaded region. Measurements were obtained from the common inlet at 8.5 m height apart from the $NO_3$ photolysis rate (35 m height on adjacent tower), wind-direction and wind-speed (16.5 m on the 128 m tower).**




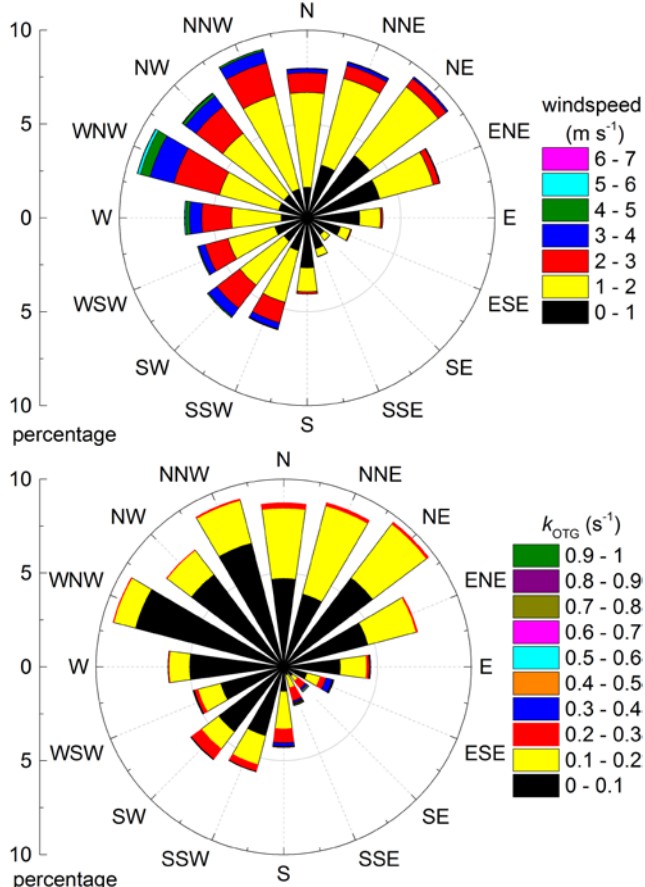

**Figure 3: Upper panel: wind rose coloured according to wind speed. Lower panel: wind rose coloured according to $k_{OTG}$.**



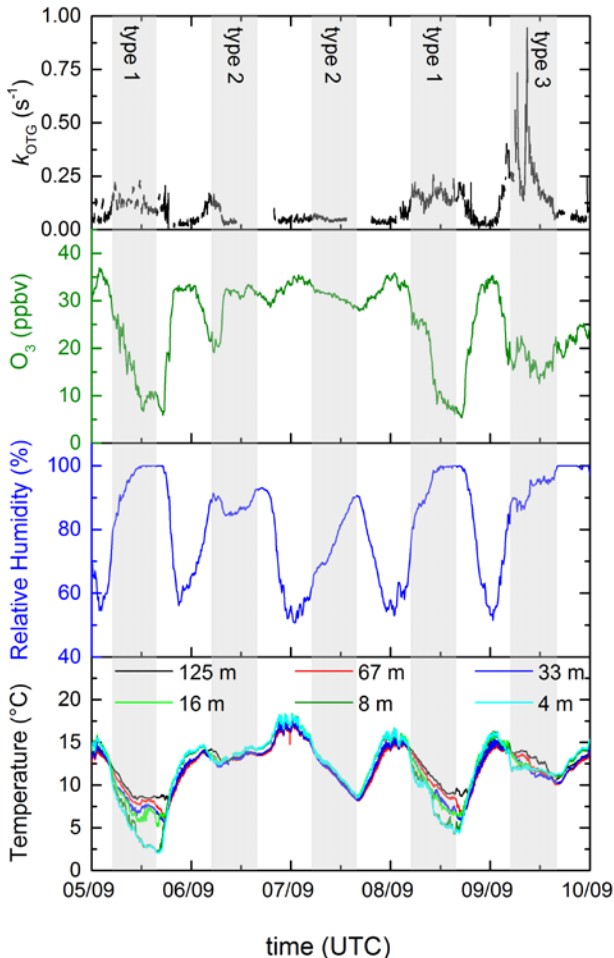

**Figure 4: Zoom-in on 5 campaign days illustrating the three types (1-3) of nighttimes encountered. Type 1 has a strong vertical gradient in temperature (lowermost panel) and significant O$_3$ loss with relative humidity at 100%. Type 2 (no temperature inversion), has little or no O$_3$ loss, and type 3 is influenced by plumes from the Korkeakoski sawmill.**





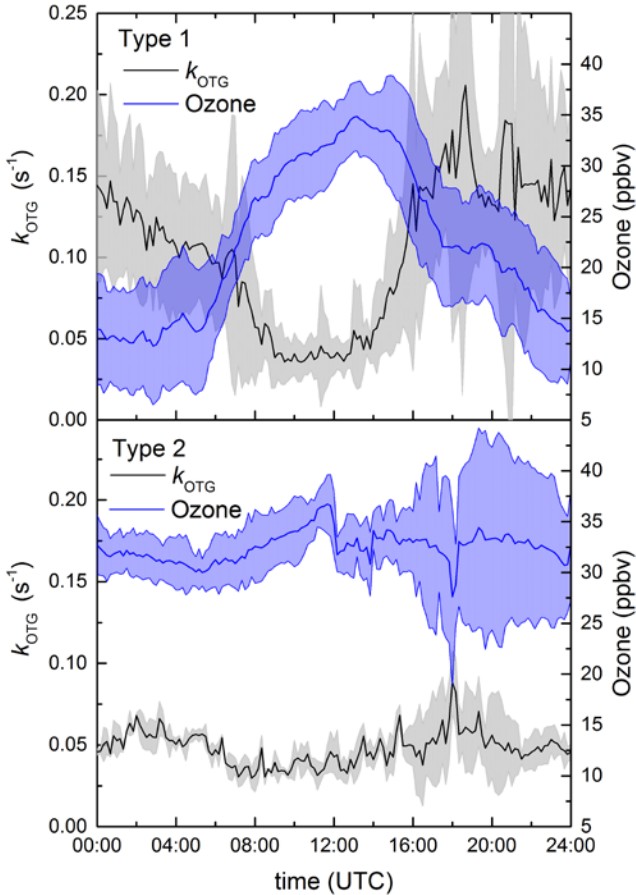

**Figure 5: Diel profiles of $k_{OTG}$ (black line) and O$_3$ (blue line) on two different types of days/nights. Upper panel: Type 1 (strong nighttime temperature inversion). Lower panel: Type 2, (no temperature inversion). The shaded areas represent 2σ uncertainty and indicate variability over the diel cycle.**



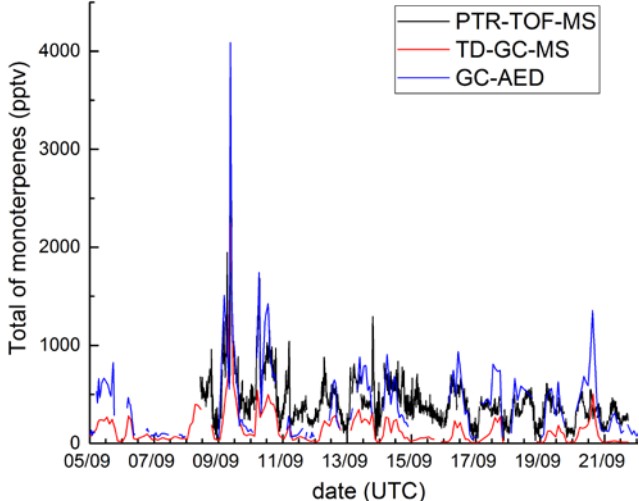

**Figure 6: Top: Time series of total mono-terpenes from GC-AED (black) GC-MS (red) and PTR-TOF-MS (blue). The data are reproduced as histograms in Fig. S2 of the supplementary information.**





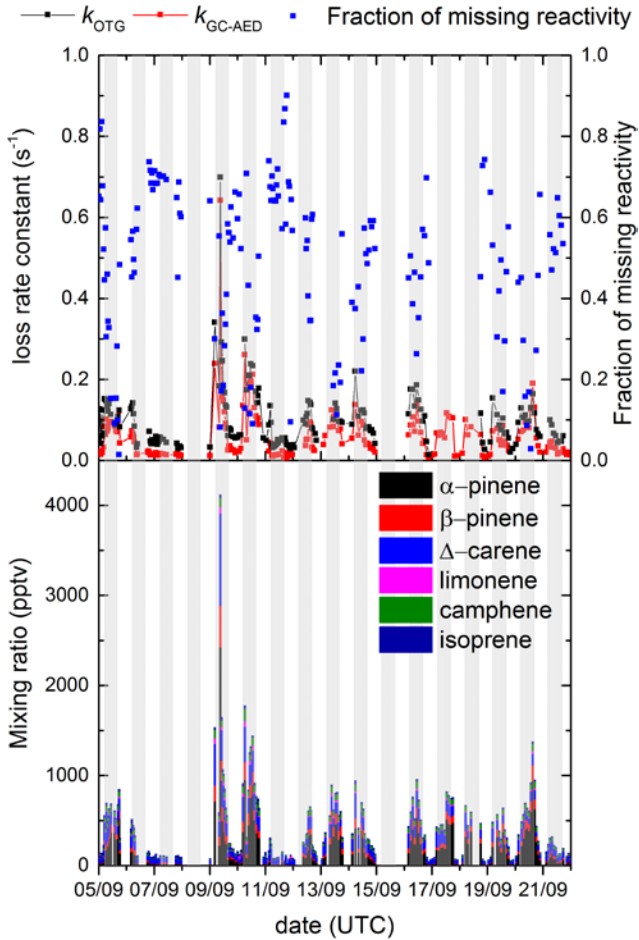

**Figure 7: Mixing ratios of individual monoterpenes as measured by the GC-AED (lower plot) and their total contribution to the NO$_3$ reactivity ($k_{GC-AED}$, red data points) compared to measured NO$_3$ reactivity ($k_{OTG}$, black data points). The fractional missing reactivity (blue data points) was calculated as ($k_{OTG}$-$k_{GC-AED}$)/$k_{OTG}$. The grey shaded areas represent nighttime.**





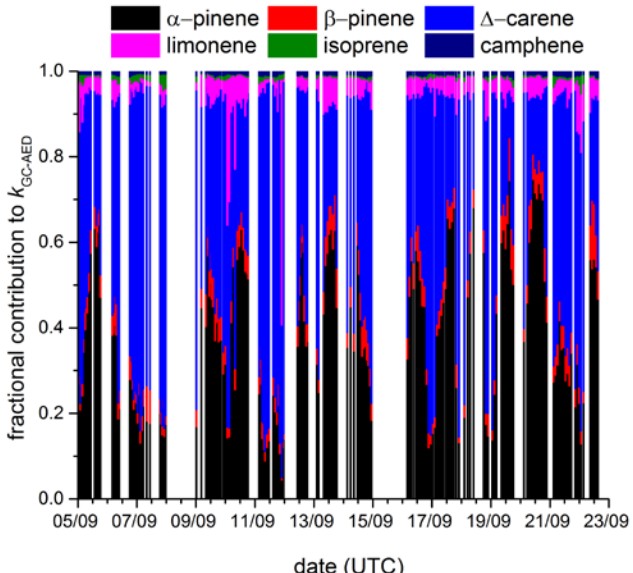

**Figure 8: Fractional contribution of individual monoterpenes (measured by the GC-AED) to $k_{\text{GC-AED}}$ indicating the dominant role of α-pinene and Δ-carene.**



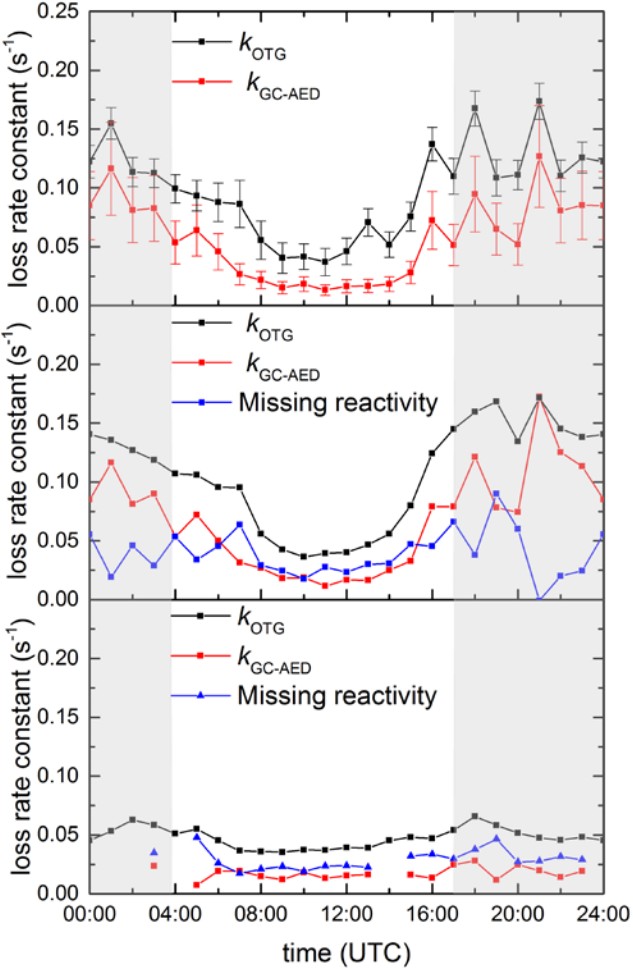

**Figure 9 Upper panel: Campaign averaged diel cycle of NO₃ reactivity ($k_{OTG}$) and the reactivity calculated from the monoterpenes reported by the GC-AED. The error bars represent the overall uncertainty in each parameter and not variability. The middle and lower panels are the values of $k_{OTG}$ and $k_{GC-AED}$ separated into different day/night types: Those with significant nighttime temperature inversion (middle) and those without (lower).**




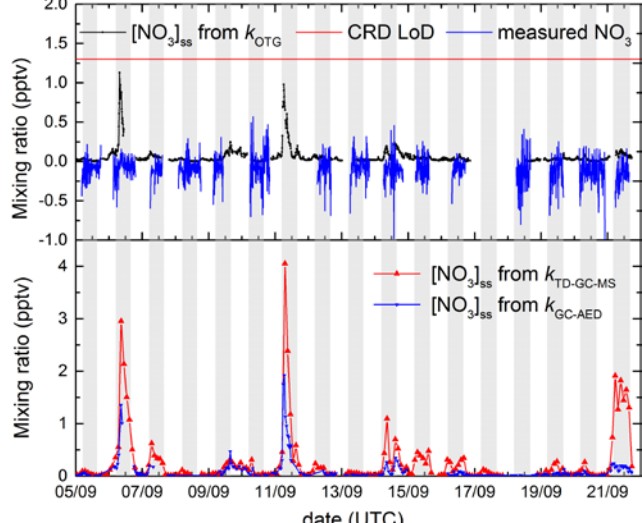

**Figure 10: Stationary state NO$_3$ mixing ratios calculated from the production term ($k_1$[NO$_2$][O$_3$]) and using either $k_{OTG}$ + $k_2$[NO] + $J_{NO3}$ (upper panel, black line), $k_{GC-MS}$ + $k_2$[NO] + $J_{NO3}$ (lower panel, red line) or $k_{GC-MS}$ + $k_2$[NO] + $J_{NO3}$ (lower panel, blue line) as loss terms. For comparison, the measured NO$_3$ mixing ratios are also plotted (upper panel, blue line) as well as the 1.3 pptv limit of detection (horizontal red line).**



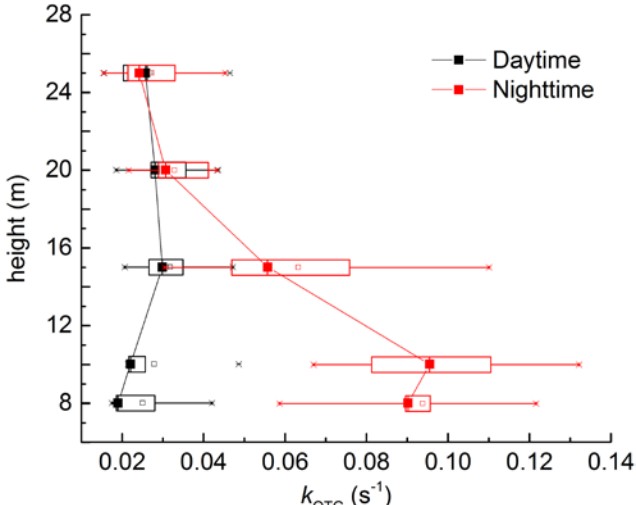

**Figure 11: Vertical profiles of NO$_3$ reactivity ($k_{OTG}$) on 17-18.09.2016. The data represent the average of 5 profiles during the day and 9 profiles during the night.**



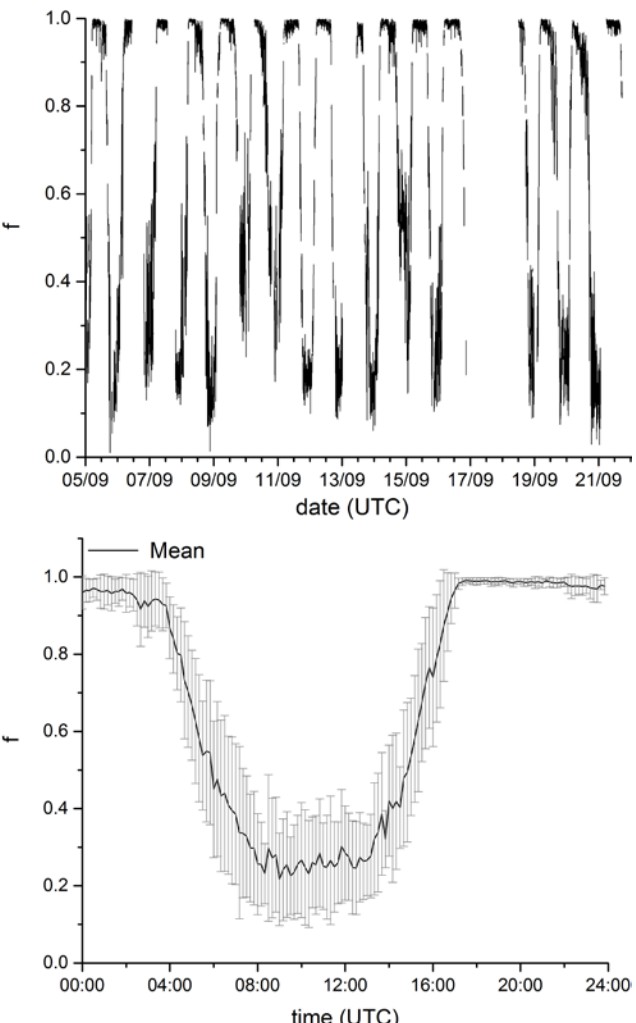

**Figure 12: The fraction, $f$, of the total NO$_3$ loss with organic trace gases as a time series (upper panel) and as a campaign averaged, diel cycle (lower panel) where $f = k_{OTG}$ / $(k_{OTG} + J_{NO3} + k_{NO})$.**