# Peer review of "Direct measurement of $\text{NO}_3$ reactivity in a boreal forest"

_Atmospheric Chemistry and Physics, 2017_

## Referee Comment (RC1) · Anonymous Referee #1 · 16 Dec 2017

General Comments:

This paper presents first measurements of NO3 reactivity in a biogenic VOC rich environment using a recently developed technique. The paper presents new results, is well written, and presents a thorough and straightforward analysis. Results for NO3 reactivity should nicely complement those for OH reactivity, which have become a standard measurement for understanding photochemistry, especially in high biogenic emitting regions. The measurement of NO3 reactivity provides a similar metric for nighttime oxidation, as well as new understanding of oxidation potential for NO3 radicals during daytime. The novelty of this result will make this paper of high interest to the audience of ACP.

The only general comment is that this measurement may pertain to a boreal forest,

but it is the first measurement of NO3 in any environment. The authors may wish to consider broadening the title somewhat to at least encompass "high biogenic emitting regions", since the results are new enough that they may generalizable beyond just a boreal forest.

Otherwise, the authors should respond to the following set of relatively minor commetns.

Specific Comments:

Page 2, line 9: check grammar

Page 2, line 26: You may want to be more specific in your definition of terpenoids, which presumably include isoprene, monoterpenes and sequiterpenes?

Page 6, line 11: calculated to be 14%, or calculated at 14%.

Page 7, line 20: Uncertainty in kOTG difficult to discern, as are values for kOTG on the linear scale given in figure 2. A log scale may be more effective for presentation.

Page 7, lines 27-30: The cited literature is generally not for forested areas. A better comparison would be to a paper such as Golz et al. (2001) or Ayres et al. (2016), which report unmeasurable NO3 in heavily forested areas.

Gölz, C., J. Senzig, and U. Platt (2001), NO3-initiated oxidation of biogenic hydrocarbons, Chemosphere - Global Change Science, 3(3), 339-352, 10.1016/s1465-9972(01)00015-0.

Ayres, B. R., H. M. Allen, D. C. Draper, S. S. Brown, R. J. Wild, J. L. Jimenez, D. A. Day, P. Campuzano-Jost, W. Hu, J. de Gouw, A. Koss, R. C. Cohen, K. C. Duffey, P. Romer, K. Baumann, E. Edgerton, S. Takahama, J. A. Thornton, B. H. Lee, F. D. Lopez-Hilfiker, C. Mohr, P. O. Wennberg, T. B. Nguyen, A. Teng, A. H. Goldstein, K. Olson, and J. L. Fry (2015), Organic nitrate aerosol formation via NO3 + biogenic volatile organic compounds in the southeastern United States, Atmos. Chem. Phys., 15(23), 13377-

13392, 10.5194/acp-15-13377-2015.

Page 8, line 6: The effect of lower wind speeds is easily tested. Were reactive VOCs also greater during these periods?

Page 8, line 12, and Figure 4 caption. "Expanded view" is a less colloquial expression than "zoom in". Minor suggestion, at authors discretion.

Page 11, line 12 and Figure 7: How well correlated are the measured and calculated $NO_3$ reactivities? A scatter plot and linear fit, possibly separated into type 1 and 2 events, would be instructive.

Page 12, line 1: Define what is meant by "statistically significant" here. Based on correlation or based on error analysis in the time series.

Page 13, lines 10-11: Have the authors considered reaction of $NO_3$ with $HO_2$ or $RO_2$? How well does the $NO_3$ reactivity instrument measure radical-radical reactions such as this, and could they contribute significantly to $NO_3$ reactivity in these environments where $NO_3$ reactivity is large?

Page 13, line 16, equation 3: Loss of $NO_3$ through either direct heterogeneous uptake or through $N_2O_5$ heterogeneous uptake is not included. This is very likely appropriate since these processes are probably slow compared to NO reaction, photolysis and VOC reaction for $NO_3$ at the SMEAR site. This should at least be mentioned.

Page 15, line 23-27: The diel cycle in the fraction of $NO_3$ reacting with VOCs is a quite useful metric and shows the relevance of $NO_3$ as a daytime oxidant. However, the $NO_3$ production rate is itself quite small at this location. Can the inferred absolute oxidaiton rate also be given (i.e., $NO_3$ production x f), and can this also be compared to similar estimates for OH or $O_3$ oxidaiton during the day?

---

## Referee Comment (RC2) · Anonymous Referee #2 · 29 Jan 2018

The manuscript by Liebmann et al. presents observations of nitrate radical reactivity together with concentrations of speciated VOC and other trace gases in a boreal forest in Finland. NO$_3$ reactivities were found to be high, especially during nights with strong surface inversions. High nocturnal stability also favored low ozone mixing ratios, likely due to O$_3$ deposition. A comparison with reactivities calculated based on the VOC observations reveal a "missing" NO$_3$ sink of 30% during the night and 60% during the day. The authors also present vertical reactivity profiles which show strong nighttime gradients with highest levels near the surface. This is a very interesting and comprehensive study, that presents unique observations and a thorough interpretation of the findings. The paper is very well written and the authors arguments are easy to follow. I found a few minor issues in the manuscript that could be clarified (see below),

[Figure]

but overall I recommend the paper for publication in ACP without major changes.

Minor Comments:

Page 8 line 10-13: Here nights are classified based on $NO_3$ reactivity. In the rest of the manuscript types 1 and 2 are typically referred to as night with and without strong surface inversions (see page 8 line 22). It would help the manuscript to stay with one definition for type 1 and 2 nights.

Page 9: I am missing a discussion of the ozone loss associated with the $NO_3$ + VOC reactions. Depending on the source of $NO_2$ (reservoir/transport vs. local NO + $O_3$ → $NO_2$), at least one ozone molecule is lost during each reaction. While this is likely not the dominant source, with sufficient reaction time of a few hours it should contribute to the ozone loss.

Figure 7: The lower panel is very difficult to read. Could it be split it up into one panel with the total mixing ratio and another panel with the fractional distribution of the BVOCs?

Figure 9: Is this average diurnal cycle determined with type 3 nights? If so what is their impact on the average?
* * *

---

## Author Comment (AC1) · 31 Jan 2018

In the following, the referee's comments are reproduced (black) along with our replies (blue) and changes made to the text (red) in the revised manuscript.

**Referee 1**

General Comments:

This paper presents first measurements of $NO_3$ reactivity in a biogenic VOC rich environment using a recently developed technique. The paper presents new results, is well written, and presents a thorough and straightforward analysis. Results for $NO_3$ reactivity should nicely complement those for OH reactivity, which have become a standard measurement for understanding photochemistry, especially in high biogenic emitting regions. The measurement of $NO_3$ reactivity provides a similar metric for nighttime oxidation, as well as new understanding of oxidation potential for $NO_3$ radicals during daytime. The novelty of this result will make this paper of high interest to the audience of ACP.

We thank referee 1 for this review and overall positive assessment of our manuscript. The manuscript has been improved in line with the comments listed below.

The only general comment is that this measurement may pertain to a boreal forest, but it is the first measurement of $NO_3$ in any environment. The authors may wish to consider broadening the title somewhat to at least encompass "high biogenic emitting regions", since the results are new enough that they may generalizable beyond just a boreal forest. Otherwise, the authors should respond to the following set of relatively minor comments.

In future publications we shall describe results from other BVOC-rich environments add thus prefer to delineate these studies by choosing more specific titles rather than generalize.

Page 2, line 9: check grammar

Corrected, we now write:

"taking place" has been replaced with "takes place".

Page 2, line 26: You may want to be more specific in your definition of terpenoids, which presumably include isoprene, monoterpenes and sesquiterpenes?

Corrected, we now write:

In forested environments at low $NO_X$ the lifetime of $NO_3$ with respect to chemical losses during the temperate months will generally be driven by the terpenoids (isoprene, monoterpenes and sesquiterpenes), the reaction proceeding via addition to the C=C double bond to form nitroxy-alkyl peroxy radicals.

Page 6, line 11: calculated to be 14%, or calculated at 14%.

Corrected

calculated to be 14%

Page 7, line 20: Uncertainty in $k_{OTG}$ difficult to discern, as are values for $k_{OTG}$ on the linear scale given in figure 2. A log scale may be more effective for presentation.

The plot has been reproduced with log-scaling in the supplementary information and this is now referred to in the caption to Figure 2.

Page 7, lines 27-30: The cited literature is generally not for forested areas. A better comparison would be to a paper such as Golz et al. (2001) or Ayres et al. (2016), which report unmeasurable $NO_3$ in heavily forested areas. Gölz, C., J. Senzig, and U. Platt (2001), $NO_3$-initiated oxidation of biogenic hydrocarbons, Chemosphere - Global Change Science, 3(3), 339-352, 10.1016/s1465-9972(01)00015-0.

Ayres, B. R., . M. Allen, D. C. Draper, S. S. Brown, R. J. Wild, J. L. Jimenez, D. A. Day, P. Campuzano-Jost, W. Hu, J. de Gouw, A. Koss, R. C. Cohen, K. C. Duffey, P. Romer, K. Baumann, E. Edgerton, S. Takahama, J. A. Thornton, B. H. Lee, F. D. Lopez-Hilfiker, C. Mohr, P. O. Wennberg, T. B. Nguyen, A. Teng, A. H. Goldstein, K. Olson, and J. L. Fry (2015), Organic nitrate aerosol formation via NO3 + biogenic volatile organic compounds in the southeastern United States, Atmos. Chem. Phys., 15(23), 13377- C2 13392, 10.5194/acp-15-13377-2015.

We have added the references and extra text:

Our short $NO_3$ lifetimes are however compatible with very low $NO_3$ mixing ratios in forested regions with high rates of emission of biogenic trace gases (Gölz et al., 2001; Rinne et al., 2012; Ayres et al., 2015).

Page 8, line 6: The effect of lower wind speeds is easily tested. Were reactive VOCs also greater during these periods?

Low wind speed alone did not necessarily result in high reactivity as the effect is convoluted with wind direction. We have illustrated this by writing:

Enhanced reactivity from the SE may be caused by emissions from the sawmill at Korkeakoski (Eerdekens et al., 2009), or a local wood-shed storing freshly cut timber about 100 m distant from the containers. This may have been compounded by the lower than average wind-speeds associated with air masses from the SE, which reduced the rate of exchange between the nocturnal boundary layer and above canopy air, effectively trapping ground-level emissions into a shallow boundary layer.

Page 8, line 12, and Figure 4 caption. "Expanded view" is a less colloquial expression than "zoom in". Minor suggestion, at authors discretion.

We now write:

Figure 4 shows an expanded view of $k_{OTG}$ ……

Page 11, line 12 and Figure 7: How well correlated are the measured and calculated $NO_3$ reactivities? A scatter plot and linear fit, possibly separated into type 1 and 2 events, would be instructive.

The correlation plot is now displayed as Fig S1 of the supplementary information. We write:

The correlation between $k_{OTG}$ and $k_{GC-AED}$ is displayed as Fig. S7 and indicates, on average that measured organics accounted for ≈ 70 % of the total $NO_3$ reactivity.

Page 12, line 1: Define what is meant by "statistically significant" here. Based on correlation or based on error analysis in the time series.

On page 11, we already wrote: "The values of $k_{OTG}$ and $k_{GC-AED}$ do not agree within their combined uncertainties, indicating that the missing reactivity calculated in equation (3) is statistically significant"

Page 13, lines 10-11: Have the authors considered reaction of $NO_3$ with $HO_2$ or $RO_2$? How well does the $NO_3$ reactivity instrument measure radical-radical reactions such as this, and could they contribute significantly to $NO_3$ reactivity in these environments where $NO_3$ reactivity is large?

Sampling peroxyl radicals would lead to a positive bias in $k_{OTG}$ when compared to in-situ measurements of VOCs but only when VOCs related reactivity is low. Our instrument will however does not measure the reactivity due to radicals such as $HO_2$ and $RO_2$ which will not survive transport through the inlets into the flow-tube. This is one reason why we name our measurement $k_{OTG}$, highlighting the fact that only reactivity due to VOCs is accessed. This appears to be unlikely for the present campaign, but may be important in assessing $NO_3$ lifetimes in less reactive air masses. We now write:

Unidentified monoterpenes / sesquiterpenes are likely to account for a significant fraction of the VOC-derived missing reactivity.

Page 13, line 16, equation 3: Loss of $NO_3$ through either direct heterogeneous uptake or through $N_2O_5$ heterogeneous uptake is not included. This is very likely appropriate since these processes are probably slow compared to NO reaction, photolysis and VOC reaction for NO3 at the SMEAR site. This should at least be mentioned.

We now write:

This expression does not consider indirect loss of $NO_3$ via heterogeneous loss processes of $N_2O_5$, which, given the high levels of BVOC (short $NO_3$ lifetimes) and low aerosol surface area, cannot contribute significantly.

Page 15, line 23-27: The diel cycle in the fraction of $NO_3$ reacting with VOCs is a quite useful metric and shows the relevance of $NO_3$ as a daytime oxidant. However, the $NO_3$ production rate is itself quite small at this location. Can the inferred absolute oxidation rate also be given (i.e., $NO_3$ production x f), and can this also be compared to similar estimates for OH or $O_3$ oxidation during the day?

This will be addressed in a separate paper from this campaign that examines the day and nighttime production of organic nitrates from the reaction of OH and $NO_3$ with BVOCs. This detailed analysis will include datasets for particle and gas-phase nitrates and its proper treatment is beyond the scope of the present manuscript.

**Referee 2**

The manuscript by Liebmann et al. presents observations of nitrate radical reactivity together with concentrations of speciated VOC and other trace gases in a boreal forest in Finland. NO3 reactivities were found to be high, especially during nights with strong surface inversions. High nocturnal stability also favored low ozone mixing ratios, likely due to $O_3$ deposition. A comparison with reactivities calculated based on the VOC observations reveal a "missing" $NO_3$ sink of 30% during the night and 60% during the day. The authors also present vertical reactivity profiles which show strong nighttime gradients with highest levels near the surface. This is a very interesting and comprehensive study that presents unique observations and a thorough interpretation of the findings. The paper is very well written and the authors arguments are easy to follow. I found a few minor issues in the manuscript that could be clarified (see below), but overall I recommend the paper for publication in ACP without major changes.

We thank referee 2 for this review and overall positive assessment of our manuscript. The manuscript has been improved in line with the comments listed below.

Minor Comments:

Page 8 line 10-13: Here nights are classified based on $NO_3$ reactivity. In the rest of the manuscript types 1 and 2 are typically referred to as night with and without strong surface inversions (see page 8 line 22). It would help the manuscript to stay with one definition for type 1 and 2 nights.

Corrected, we now write:

In order to examine the difference in daytime and nighttime $NO_3$ reactivity and also to explain the large nighttime variability in $k_{OTG}$ we categorize the nights into three broad types: 1) nights with strong temperature inversion where the $NO_3$-reactivity was greatly increased compared to the previous or following day, 2) nights without temperature inversion with comparable (usually low) daytime and nighttime $NO_3$-reactivity, and 3) events with unusually high $NO_3$-reactivity.

Page 9: I am missing a discussion of the ozone loss associated with the $NO_3 + VOC$ reactions. Depending on the source of $NO_2$ (reservoir/transport vs. local $NO + O_3! NO_2$), at least one ozone molecule is lost during each reaction. While this is likely not the dominant source, with sufficient reaction time of a few hours it should contribute to the ozone loss.

The nocturnal loss of $O_3$ due to reaction with $NO_2$ is not significant. Even if we take the maximum observed $NO_2$ concentration at night (2 ppbv) and assume that all $NO_3$ reacts with VOCs (i.e. no reformation of NOx) we calculate that less than 1 ppbv $O_3$ will be converted (via reaction with $NO_2$) in 4 hours. To clarify this we write:

O$_3$ depletion due to its slow reaction with NO$_2$ (present at maximum 2 ppbv at night) does not contribute significantly to its loss even if all resultant NO$_3$ reacts to form organic nitrates rather than form N$_2$O$_5$ and re-release NOx.

Figure 7: The lower panel is very difficult to read. Could it be split it up into one panel with the total mixing ratio and another panel with the fractional distribution of the BVOCs?

Splitting the lower panel into total mixing ratio and fractional contribution would not make it more legible. The summed mixing ratios of the monoterpenes are already displayed in Figure 6.

Figure 9: Is this average diurnal cycle determined with type 3 nights? If so what is their impact on the average?

The average diurnal cycle (upper panel) includes the "type 3" nights. Exclusion of these nights does however not have a significant impact on the mean as the two saw-mill events (over 17 days total) were short lived.

We have re-labelled the plot to better identify the types of nights and clarified the inclusion of type 3 in the figure caption.